# Tofacitinib Treatment Suppresses CD4+ T-Cell Activation and Th1 Response, Contributing to Protection against Staphylococcal Toxic Shock

**DOI:** 10.3390/ijms25137456

**Published:** 2024-07-07

**Authors:** Anders Jarneborn, Zhicheng Hu, Meghshree Deshmukh, Pradeep Kumar Kopparapu, Tao Jin

**Affiliations:** 1Department of Rheumatology and Inflammation Research, Institute of Medicine, Sahlgrenska Academy, University of Gothenburg, 413 46 Gothenburg, Sweden; anders.jarneborn@vgregion.se (A.J.); zhicheng.hu@gu.se (Z.H.); meghshree.vinod.deshmukh@gu.se (M.D.); pradeep.kopparapu@gu.se (P.K.K.); 2Department of Rheumatology, Sahlgrenska University Hospital, 413 45 Gothenburg, Sweden

**Keywords:** staphylococcal toxic shock syndrome, mouse, tofacitinib, Th1 response

## Abstract

Staphylococcal toxic shock syndrome (STSS) is a rare, yet potentially fatal disease caused by *Staphylococcus aureus* (*S. aureus*) enterotoxins, known as superantigens, which trigger an intense immune response. Our previous study demonstrated the protective effect of tofacitinib against murine toxin-induced shock and a beneficial effect against *S. aureus* sepsis. In the current study, we examined the effects of tofacitinib on T-cell response in peripheral blood using a mouse model of enterotoxin-induced shock. Our data revealed that tofacitinib suppresses the activation of both CD4+ and CD8+ T cells in peripheral blood. Furthermore, both gene and protein levels of Th1 cytokines were downregulated by tofacitinib treatment in mice with enterotoxin-induced shock. Importantly, we demonstrated that CD4+ cells, but not CD8+ cells, are pathogenic in mice with enterotoxin-induced shock. In conclusion, our findings suggest that tofacitinib treatment suppresses CD4+ T-cell activation and Th1 response, thereby aiding in protection against staphylococcal toxic shock in mice. This insight may guide the future development of novel therapies for STSS.

## 1. Introduction

Staphylococcal toxic shock syndrome (STSS) is a rare yet potentially lethal *Staphylococcus aureus (S. aureus)*-disease, triggered by *S. aureus* toxins, known as superantigens, which provoke an intense immune response [1,2]. These toxins, including toxic shock syndrome toxin-1 (TSST-1), staphylococcal enterotoxins, and staphylococcus endotoxin-like toxins [3], bind TCR through the Vβ-chain with MHC class II molecules, inducing a profound inflammatory response through massive polyclonal activation of T cells [4]. Other infectious agents, such as streptococcus, are also known to produce superantigens. STSS initially gained attention through menstrual TSS, observed in young women experiencing severe toxic shock linked to tampon use, creating an environment favourable for TSST-1-producing *S. aureus*. Beside the fulminant TSS, these toxins are also known to be virulence factors for other types of *S. aureus* infections, such as septic arthritis [5]. STSS is a severe, dramatic condition with high mortality. Treatment includes critical care organ support, empirical antibiotic treatment, and potentially intravenous immunoglobulin treatment to modulate the immune response [6].

Considering the potential ability of the superantigens to hyperactivate the immune system in an uncontrolled manner, treatment aimed at modulating this exaggerated response towards a more balanced state holds significant promise, both in toxic shock syndrome and in the dysregulated immune response seen in sepsis. In sepsis, the prevailing understanding is that an excessive pro-inflammatory reaction is followed by an immunosuppressive phase, which contributes significantly to mortality. Despite numerous attempts to use immunosuppressive agents in sepsis, few have yielded positive results in human trials, likely due to the diverse underlying causes of sepsis and septic shock, as well as the absence of robust biomarkers for immune status [7]. 

Janus kinase inhibitors (JAKi) represent a modern class of immunomodulating drugs in treating rheumatic disease, inflammatory bowel disease, skin disorders, and haematological conditions [8]. They function by inhibiting Janus kinases (JAKs), a group of kinases associated with numerous surface receptors for a vast number of cytokines and hormones [9]. Upon ligand binding to their receptors, the JAKs (including JAK1, JAK2, JAK3, and TYK2) phosphorylate intracellular proteins known as signal transducers and activators of transcription (STAT). These phosphorylated STAT proteins then dimerise and translocate to the nucleus, initiating gene transcription [10]. With their broad effect on immune signalling, reasonable half-life, and reversible binding, these drugs have potential for immunomodulation in cases of severe immune dysregulation triggered by infection and bacterial toxins. JAKi have been shown to have a protective effect in some infections, such as polymicrobial and fungal sepsis [11,12].

Tofacitinib, one of the initial JAKi, is approved for rheumatoid arthritis; it exhibits a higher affinity for JAK1 and JAK3. During the COVID-19 pandemic, it demonstrated effectiveness in severe COVID-19-related hyperinflammation in humans [13]. Our group has previously shown the remarkable protective effect of tofacitinib against murine toxin-induced shock and a positive impact on *S. aureus* sepsis caused by a superantigen-producing strain [14]. In the current study we investigated the alterations in T-cell response in peripheral blood by tofacitinib in a mouse model for enterotoxin-induced shock. Our aim was to potentially identify key markers to determine when tofacitinib could be used to mitigate or prevent severe shock in patients.

## 2. Results

### 2.1. Tofacitinib Has a Limited Effect on Mitigating the Loss of Peripheral T Cells Caused by Toxic Shock

To study the role of T cells in the protective effect of tofacitinib in toxin-induced shock, blood was collected from toxin-challenged mice, treated with tofacitinib or vehicle only, and analysed by FACS (Figure 1a). Untreated healthy controls were used for reference. Toxin challenge led to a severe decrease in CD3+ cells in blood compared to levels in healthy mice. Levels between toxin-challenged mice treated with tofacitinib and controls were not significantly different. (Figure 1b). Further characterisation showed that the ratio of CD4+ out of all T cells was higher in tofacitinib-treated mice than in mice treated with vehicle only (Figure 1c), while, conversely, CD8+ cells showed a lower ratio in the tofacitinib group (Figure 1d). 

### 2.2. Tofacitinib Reduces Activation of T Cells in Peripheral Blood

TSST-1 is a superantigen, with the ability to induce a massive activation of T cells [15]. While the number of circulating T cells was not significantly different between tofacitinib and vehicle-treated mice, further analysis of T-cell activation level was conducted. Using CD69+ as an activation marker, we found that in the mice receiving tofacitinib the activated proportion of both CD4+ and CD8+ cells was significantly reduced (Figure 2a–c). Consequently, while tofacitinib shows only a slight ability to limit the reduction of circulating T cells in toxic shock, its ability to inhibit activation is pronounced.

### 2.3. Tofacitinib Reduces Th1 Cytokine Expression in Mice with Toxic Shock

To further examine the effect of tofacitinib on the T-cell environment in peripheral blood, gene expression was analysed. As cytokines typical for a Th1 response, *Il-12* and *Ifn-γ* were chosen, and, as a marker of a Th2 response, *Il-4* was used [16]. Blood was collected from mice challenged and treated as described above, and mRNA expression was analysed by qPCR. Tofacitinib-treated mice showed no significant difference in expression of *Il-12* or *Ifn-γ* compared to healthy mice, while toxin-challenged control mice showed significantly upregulated expression in these genes (Figure 3a,b). *Il-12* expression was significantly higher in toxin-challenged controls compared to mice with tofacitinib treatment, while *Ifn-γ* showed a trend towards downregulation by tofacitinib. Expression of the prototypic Th2 cytokine *Il-4*, while highly expressed in healthy Balb/c mice, was downregulated in both treatment groups without significant difference (Figure 3c). These findings strongly suggest that the Th1 response pattern is most affected by tofacitinib.

### 2.4. Tofacitinib Reduces Blood Levels of IL-12, IFN-γ, and IL-6 in Mice with Toxic Shock

To correlate gene expression data with actual protein levels in blood, ELISA was used to detect the cytokine levels. Similarly to what was seen in previously examined mRNA expression, the level of IL-12 was significantly lower in tofacitinib-treated animals compared to toxin-challenged controls (Figure 4a), while IFN-γ showed a strong trend in the same direction (Figure 4b). Furthermore, the important pro-inflammatory cytokine IL-6 showed a trend towards lower levels (Figure 4c). This confirms the findings in mRNA expression and shows that not only are the genes beginning to be expressed at this time point of shock, but a difference in protein levels is also noticeable.

### 2.5. CD4+ but Not CD8+ Cells Are Pathogenic in Mice with Toxin-Induced Shock

To further understand the role of T cells and what cells are most important in toxin-induced shock, we depleted CD4+ cells and CD8+ cells, respectively, in the toxic shock model. Balb/c mice received depleting antibodies targeting CD4, CD8, or an isotype control. Mice were then challenged with toxins and monitored for survival. Mice depleted of peripheral CD8 cells showed identical survival to controls, with all succumbing to shock within 27 h (Figure 5). Among CD4-depleted mice, however, the first death came after 47 h, and 20% survived the experiment, suggesting that the key pathogenic T cells in the toxin-induced shock model are the CD4+ T cells. Mice surviving the acute phase of shock in this model show complete clinical recovery and live on without any clinical signs of discomfort or complications.

## 3. Discussion

In this study, we demonstrate the pivotal role of CD4+ cells in the pathogenesis of enterotoxin-induced shock. Our previous findings have revealed the potent protective effect of the JAK inhibitor tofacitinib in mice challenged with staphylococcal enterotoxins, with part of this protection attributed to the inhibition of CD4+ cell activation.

The main effect of superantigens, including TSST-1, is their potent capacity to activate up to 20% of T cells, triggering a massive inflammatory response, in contrast to 0.01% activation in a regular antigen-dependent manner [1]. T cells have been identified as key players in enterotoxin-induced shock. Previous studies have underscored the protective potential against toxic shock when T cells are targeted. For instance, experiments involving the reintroduction of T cells in SCID (severe combined immunodeficiency) mice have demonstrated a significant mitigation of shock symptoms. Similarly, treatment with ciclosporin, known for its immunosuppressive properties by specifically targeting T cells, has shown promise in attenuating the severity of enterotoxin-induced shock [17]. Moreover, CTLA4Ig, a fusion protein that inhibits the co-stimulatory signal necessary for T-cell activation, has exhibited notable protective effects in the management of enterotoxin-induced shock in an experimental setting [18]. In our study, we observed a significant reduction in peripheral blood T-cell levels following toxin challenge in mice. This decrease suggests that despite a notable expansion of T cells in response to the enterotoxin challenge, their numbers in the bloodstream diminish, likely due to extensive migration of activated T cells into vital organs or tissues. Indeed, sepsis and septic shock are known to potentially induce lymphopenia, which can be associated with a worsened outcome [19]. Tofacitinib, known for its effects on T-cell activation and differentiation [20], was administered to the mice with enterotoxin-induced shock in our study. Importantly, treated mice exhibited a significantly lower proportion of activated T cells in blood, indicating that the initial robust activation is impeded. Moreover, it is probable that tofacitinib also reduces the presence of activated T cells in affected tissues, which should be further studied.

Our data strongly indicate that CD4+ T cells, rather than CD8+ T cells, play a pathogenic role in mice with enterotoxin-induced shock. We observed that depletion of CD4+ T cells, but not CD8+ T cells, rescued 20% of mice from lethal toxin-induced shock. These findings are consistent with evidence from the endotoxin-induced shock model. Transfer of CD4+CD25– T cells to RAG−/− mice lacking T and B cells before endotoxin challenge significantly exacerbated shock parameters, including a notable increase in TNF-α and IFN-γ release, as well as a pronounced decrease in temperature and blood glucose levels compared to controls. This underscores the crucial involvement of CD4+ T cells in toxin-induced shock [21]. Interestingly, co-transfer of CD4+CD25+ natural regulatory T cells (Tregs) with CD4+CD25– T cells inhibited the expansion of IFN-γ-producing T cells and reduced endotoxin responses [21]. This suggests that Tregs, a subpopulation of CD4+ T cells, have a protective rather than pathogenic role in this context.

It is important to note that there was not full protection (only a 20% survival rate) in the CD4-depletion group. This suggests that while CD4+ T-cell activation is a significant contributing factor, other contributors must be acting synergistically with CD4 T cells. For example, TNF-α is known to be crucial in the enterotoxin-induced model as TNF Rp55−/− mice are resistant to toxin challenge [22]. Monocytes are the major producers of TNF-α, and it is very likely that monocyte activation also contributes to lethal shock. Tofacitinib may inhibit monocyte activation as well. The combined inhibition of CD4 T cells and monocyte activation could explain the full protective effect observed in the tofacitinib pre-treatment experiment [14]. While our current study focuses primarily on understanding the role of T-cell populations in lethal shock, we plan to explore the role of monocyte activation and its inhibition by tofacitinib in future studies. 

Although not solely accountable for the pathology of toxic shock, an early Th1 response plays a pivotal role in the disease’s dramatic course [23,24]. Our study shows reduced levels of IFN-γ and the crucial Th1 cytokine IL-12 in tofacitinib-treated septic shock mice. IL-12 is involved in the differentiation of naive T cells into Th1 CD4+ T cells. IFN-γ, the critical Th1 cytokine, is primarily produced by activated T lymphocytes and natural killer cells and orchestrates a multifaceted response against pathogens and tumours [25]. Importantly, IFN-γ has also been shown to play a pivotal role in the enterotoxin-induced shock model as treatment with IFN-γ-neutralising antibodies partially ameliorates lethality [17,25]. This underscores the intricate balance between host immune defence and immunopathology as dysregulated IFN-γ signalling can exacerbate tissue damage and contribute to the progression of inflammatory diseases. 

Several agents have demonstrated potential in animal studies for combating the hyperinflammation associated with toxic shock and sepsis [26,27]. However, none has transitioned into clinical practice [28,29]. This could be attributed to the heterogeneity of causative agents and comorbidities among patients compared to the controlled conditions of animal models. Additionally, the dynamic interplay between hyperinflammation and immunosuppression in the context of severe sepsis and shock could also contribute to this challenge. The current methods for assessing a patient’s immune status are very limited, with few, if any, bedside tests available to aid clinicians in determining whether immune suppression or immune stimulation is warranted. The development of such tools would be greatly beneficial, allowing clinicians to guide treatment more accurately towards advanced and, above all, safe immunomodulation strategies [7]. The primary aim of the present study was to examine alterations induced by enterotoxins in peripheral blood in mice, given its accessibility in a clinical setting. Our findings indicate that CD4+ T-cell activation may represent a significant aspect, yet further investigation is warranted to evaluate the relevance of other cell types, particularly monocytes and B cells. 

What is the clinical implication of our findings? In the context of COVID-19 hyperinflammation, several agents, including the anti-IL-6 monoclonal antibodies tocilizumab and dexamethasone, have demonstrated efficacy and are being utilised in clinical practice [30,31]. While many biologics tested in sepsis have extended elimination half-lives, potentially posing challenges in the later stages of immunosuppression in sepsis and septic shock, the ideal agent would have a shorter half-time in humans. Obviously, tofacitinib is a more flexible option compared with anti-IL-6 antibodies due to its shorter half-life pharmacokinetics. When coupled with peripheral blood markers to assess and monitor crucial components of immune status, such an immunosuppressive agent could be administered early in the disease course when a pro-inflammatory pathology predominates. It could then be discontinued once the cytokine storm has abated, facilitating recovery and potentially mitigating the severity of subsequent immunosuppression.

## 4. Materials and Methods

### 4.1. Mice

Female Balb/c mice, aged 6 weeks, were ordered from Envigo (Venray, The Netherlands) and kept at the animal facilities of the University of Gothenburg. Mice were housed under standard light and temperature conditions and fed laboratory chow and water ad libitum. The Ethics Committee of Animal Research of Gothenburg approved the study, and the guidelines for animal experimentation of the Swedish Board of Agriculture were strictly adhered to. 

### 4.2. Mouse Model of Toxin-Induced Shock

To induce a state of toxic shock, we used an established mouse model utilising two toxins, the *Staphylococcus aureus (S. aureus)*-derived toxin toxic shock syndrome toxin-1 (TSST-1) and lipopolysaccharide (LPS) [24,32]. Female Balb/c mice were challenged intra-peritoneally (i.p.) with 10 μg TSST-1 (Sigma Aldrich, Saint Louis, MI, USA) followed by an i.p. injection of 170 μg LPS (Sigma Aldrich) after 4 h (Figure 6). Neither of these doses are lethal on their own but have a synergistic effect, inducing a lethal state of shock. Sixteen hours after TSST-1 injection, mice were anaesthetised with an i.p. injection of medetomidine and ketamine and subsequently sacrificed for blood collection. 

### 4.3. Treatment with Tofacitinib

To study the effect of the JAK-inhibitor tofacitinib in toxin-induced shock, mice were divided into two groups, receiving either 50 mg/kg tofacitinib (Hölzel Diagnistika Handels GmbH, Köln, Germany) dissolved in 25% DMSO in PBS or only vehicle containing 25% DMSO in PBS. Drug or vehicle was administered by sub-cutaneous (sc) injection of 0.2 mL twice: three hours before and at the time of toxin administration. Tofacitinib treatment experiments in toxin-challenged mice were performed twice.

### 4.4. T-Cell Depletion

To evaluate the importance of T helper cells and cytotoxic T cells in the toxic shock model, depleting antibodies against CD4 and CD8 were used. Rat anti-mouse CD4 mAb (clone GK1.5; BioXCell, Lebanon, ME, USA) and rat anti-mouse CD8α (clone 2.43; BioXCell, Lebanon, ME, USA) were used as previously described [33]. A rat IgG2a isotype control (clone 2A3; BioXCell, Lebanon, ME, USA) was used for control. Mice were divided into three groups (*n* = 5 per group), receiving either anti-CD4, anti-CD8α, or isotype control. Antibodies were administered by i.p. injection with a dose of 400 μg dissolved in 200 μL of PBS. Injections were given one day prior to toxin challenge and again 4 h before start of toxin challenge. Toxic shock was induced as described above. Mice were monitored for mortality at a minimum of 4 times per 24 h and at least every 8 h. Mice were sacrificed if showing signs of severe symptoms of shock (rugged fur, isolation from group, immobility). If deemed too ill to survive until the next time point, the mouse was sacrificed and considered dead due to shock.

### 4.5. Flow Cytometry

To study the effect of tofacitinib on circulating T cells and their activation in toxin-induced shock, mouse peripheral blood was analysed with fluorescence-activated cell sorting (FACS) by using a TruCount^TM^ assay (BD Biosciences, Franklin Lakes, NJ, USA) or regular FACS tubes. Blood was collected in ethylenediaminetetraacetic acid tubes from mice 16 h after toxin challenge. Whole blood was transferred to a Trucount^TM^ tube and incubated for 10 min with Fc Block (BD Biosciences) to mitigate unspecific binding, before staining with antibody. Red blood cells were then lysed by adding Lysis Buffer (Invitrogen, Waltham, MA, USA) and incubated for 15 min. Alternatively, whole blood was first lysed using Lysis Buffer and washed with FACS buffer. Samples were then blocked with Fc Block for 10 min, followed by antibody staining and incubation for 30 min. All samples were acquired using a BD FACSLyric flow cytometer (BD Biosciences) and analysed using FlowJo software (version 10.10; Tree Star, Ashland, OH, USA). The following antibodies were used: BV421-conjugated anti-CD3 (BioLegend, San Diego, CA, USA), PerCP-Cy5.5-conjugated anti-CD3 (BioLegend), APC-conjugated anti-CD4 (BioLegend), PE-conjugated anti-CD8a (eBioscience, San Diego, CA, USA), FITC-conjugated anti-CD69 (Invitrogen, Waltham, MA, USA), BV510 anti-CD19 (BioLegend), PE-Cy7-conjugated anti-Ly6G (BD Bioscience), BV605-conjugated anti-Ly6C (BD Bioscience), and Fixable Viability Dye 780 (eBioscience). Gating strategy is presented in Appendix A.

### 4.6. RNA Extraction, cDNA Synthesis, and qRT-PCR 

Whole blood from mice challenged with toxins was collected and stored in Qiazol Lysis Reagent (Qiagen, Hilden, Germany). RNA was extracted using the miRNeasy Mini Kit (Qiagen, Hilden, Germany) according to manufacturer’s instructions. Quality and quantity of isolated RNA was determined using a NanoDrop 2000 Spectrophotometer (Thermo Fisher Scientific, Waltham, MA, USA). cDNA synthesis was carried out using a Superscript III First-Strand Synthesis Supermix kit (Invitrogen). The expression levels of *Ifn-γ, Il-4, Il-12a, Cd4,* and *Cd8a* were analysed by quantitative RT-PCR using Power SYBR Green gene expression assays (Applied Biosystems, Warrington, UK), and mouse *β-actin* was used as an internal control. Predesigned KiCqStart primers (KiCqStart SYBR Green, Merck, Darmstadt, Germany) were used: *Ifn-γ* (FP 5’TGAGTATTCCCAAGTTTGAG 3′) (RP 5′ CTTATTGGGACAATCTCTTCC 3′), *Il-4* (FP 5′ CTGGATTCATCGATAAGCTG 3′) (RP 5′ TTTGCATGATGCTCTTTAGG 3′)*, Il-12a* (FP 5′ GAAGACATCGATCATGAAGAC 3′) (RP 5′ CTCTTGTTGTGGAAGAAGTC 3′)*, Cd4* (FP 5′ TAGCAACTCTAAGGTCTCTAAC 3′) (RP 5′ GATAGCTGTGCTCTGAAAAC 3′), *Cd8a* (FP 5′ ATAGTACGTTCTCACCCTG 3′) (RP 5′ GAGTTCACTTTCTGAAGGAC 3′)*,* and *β-actin* (FP 5′ GATGTATGAAGGCTTTGGTC 3′) (RP 5′ TGTGCACTTTTATTGGTCTC 3′). Differences in expression were calculated using the ΔCt method.

### 4.7. Enzyme-Linked Immunosorbent Assay 

To examine levels of cytokines IL-6, IFN-γ, and IL-12 in peripheral blood of mice challenged with toxins, whole blood was collected and centrifuged at 1200 rpm for 15 min. Serum was extracted and analysed using enzyme-linked immunosorbent assay DuoSET kits (R&D Systems Europe, Abingdon, UK) according to manufacturer’s instructions.

### 4.8. Statistical Analysis

Data were analysed using GraphPad Prism v 9 (GraphPad software, La Jolla, CA, USA). Results are presented as mean ± standard error of the mean (SEM) unless otherwise stated. A result of *p* < 0.05 is considered statistically significant. The Mann–Whitney U test was used to compare cytokine levels in blood. Kruskal–Wallis test, followed by Dunn’s multiple comparative test or one-way ANOVA, followed by Tukey’s test was used to compare subpopulations of T cells and mRNA expression. The log-rank (Mantel–Cox) test was used to analyse mortality data in T cell depletion experiments. 

## Figures and Tables

**Figure 1 ijms-25-07456-f001:**
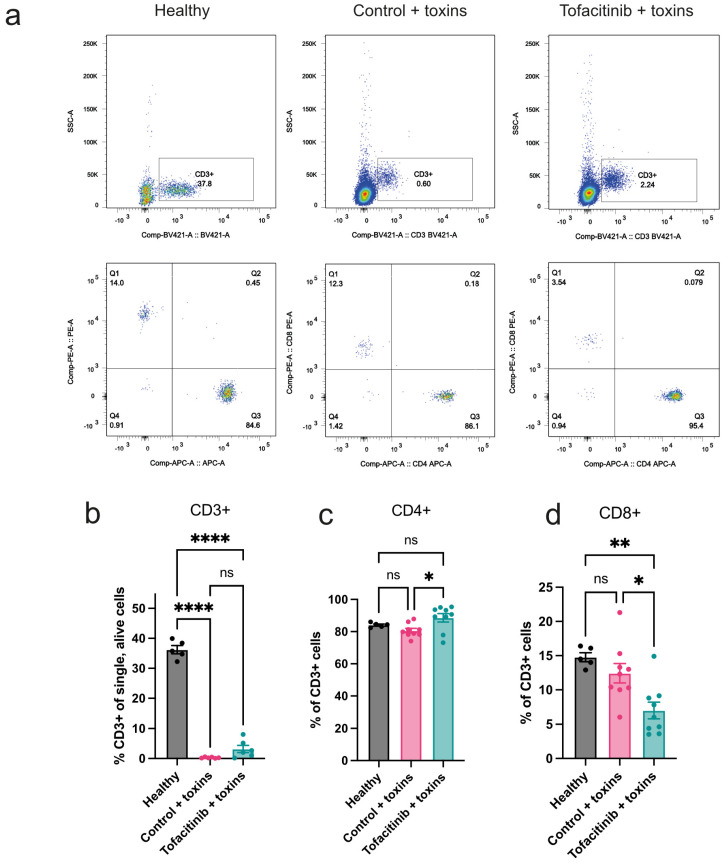
Tofacitinib has a limited effect on mitigating the loss of peripheral T cells caused by toxic shock. Balb/c mice were divided into groups, given either tofacitinib (*n* = 9) or vehicle only as control (*n* = 9), and then challenged with toxins. Untreated mice (*n* = 5) that were not given any toxins were used for reference. (**a**) Representative examples from fluorescence-activated cell sorting (FACS) analysis of one experiment showing decrease in CD3+ cells in peripheral blood following toxic challenge compared to healthy mice. (**b**) Frequency of T cells (% CD3+ of alive, single blood cells). (**c**) Frequency of T helper cells (% CD4+CD8− out of all CD3+). (**d**) Frequency of cytotoxic T cells (% CD8+CD4− of all CD3+). Frequency data were pooled from two experiments. Results are presented as mean ± standard error of the mean (SEM). *p* values were determined using one-way analysis of variance (ANOVA) with Tukey’s multiple comparison test. * *p* < 0.05; ** *p* < 0.01; **** *p* < 0.0001; ns: not significant.

**Figure 2 ijms-25-07456-f002:**
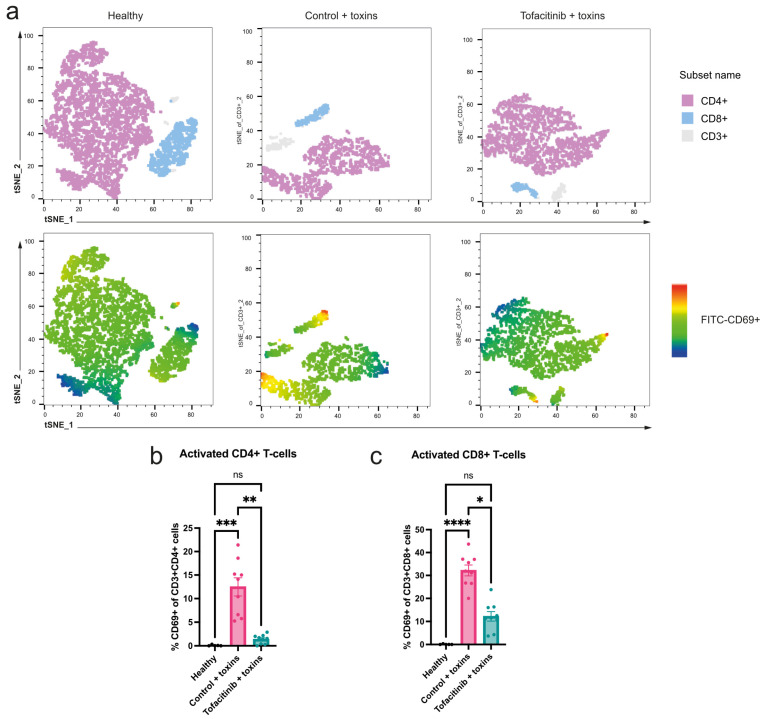
Tofacitinib reduces activation of T cells found in peripheral blood. (**a**) Top shows a t-distributed stochastic neighbour embedding (tSNE) plot of all CD3+ (grey), with CD8+ (blue) and CD4+ (purple) positive cells marked as subgroups, from peripheral blood of mice treated with either tofacitinib (*n* = 9) or vehicle only as control (*n* = 9) and then challenged with toxins; completely healthy mice (*n* = 5) were used as reference. Below are the same populations with heat mapping of CD69 expression levels in the T-cell subgroups, showing almost no activation in healthy mice, very little activation in tofacitinib-treated mice, and a clear larger proportion of the cells activated in the control mice challenged with toxins. (**b**) Frequency of T helper cells (% CD4+CD69+ of all alive CD3+). (**c**) Frequency of activated cytotoxic T cells (% CD8+CD69+ of all alive CD3+ cells). tSNE plots created in FlowJo v 10.10. Results are presented as mean ± standard error of the mean (SEM). *p* values were determined using the Kruskal–Wallis test followed by Dunn’s multiple comparative test. * *p* < 0.05; ** *p* < 0.01; *** *p* < 0.001; *****p* < 0.0001; ns: not significant.

**Figure 3 ijms-25-07456-f003:**
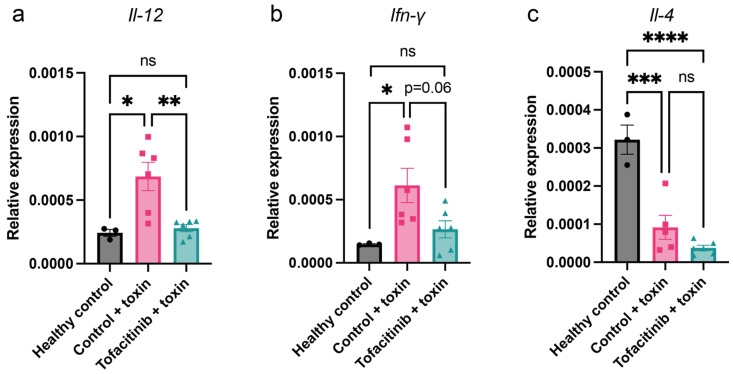
Tofacitinib reduces gene expression of key Th1 cytokines in mice with toxic shock. Blood mRNA levels of *Il-12* (**a**), *Ifn-γ* (**b**), and *Il-4* (**c**) from mice challenged with toxins and treated with either tofacitinib (*n* = 6) or vehicle only (*n* = 6) or healthy mice (*n* = 3) for reference. Results are presented as mean ± standard error of the mean (SEM). *p* values were determined using one-way ANOVA followed by Tukey’s test. * *p* < 0.05; ** *p* < 0.01; *** *p* < 0.001; **** *p* < 0.0001; ns: not significant.

**Figure 4 ijms-25-07456-f004:**
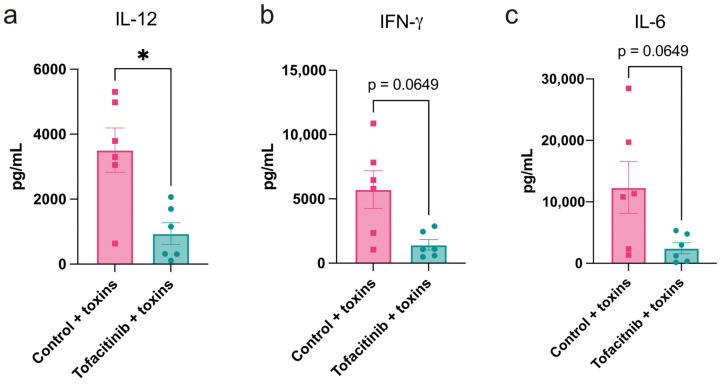
Plasma levels of key pro-inflammatory cytokines are reduced by tofacitinib in mice with toxic shock. Plasma levels of IL-12 (**a**), IFN-γ (**b**), and IL-6 (**c**) in mice following the challenge with toxic shock and pre-treatment with either tofacitinib (*n* = 6) or vehicle only (*n* = 6). Cytokine levels were non-detectable in healthy controls and are not shown. Results are presented as mean ± standard error of the mean (SEM). *p* values were determined using the Mann–Whitney U test. * *p* < 0.05; ns: not significant.

**Figure 5 ijms-25-07456-f005:**
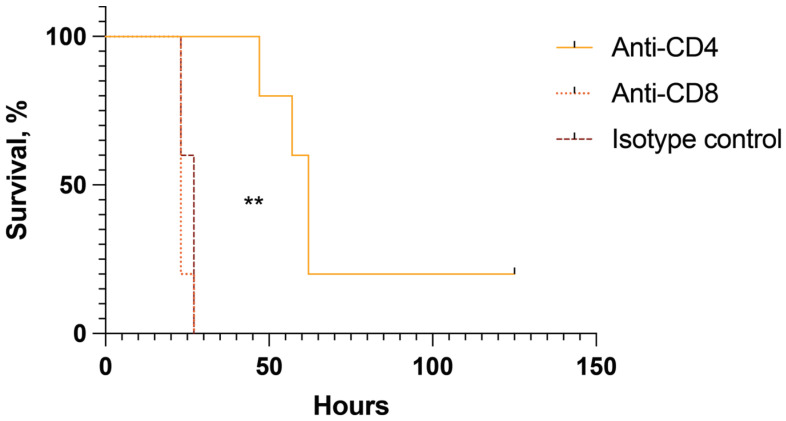
T helper cells are the main culprits among T cells in the pathogenicity of toxic shock in mice. Survival of mice challenged with toxic shock following administration of depleting antibodies against CD4 (*n* = 5), CD8 (*n* = 5), or isotype control (*n* = 5). *p* value was determined using the log-rank (Mantel–Cox) test. ** *p* < 0.01. ** indicates the difference between the CD4 group and both other groups.

**Figure 6 ijms-25-07456-f006:**
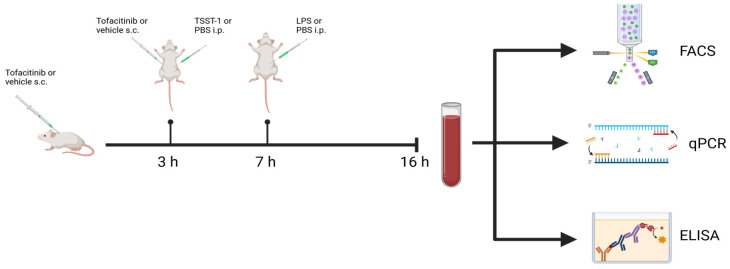
Schematic of in vivo mouse model of toxin-induced shock. Balb/c mice were treated sc. with 200 μL of either tofacitinib (50 mg/kg/injection) or vehicle only (25% DMSO in PBS). Treatment was repeated after 3 h, and at this time point, toxin challenge was initiated with a 200 μL i.p. injection of TSST-1 (10 μg/mouse). 4 h later, mice were given a 200 μL i.p. injection of LPS (170 μg/mouse). 16 h after the TSST-1 injection, mice were sacrificed for blood collection for subsequent analysis by FACS, qRT-PCR, and ELISA.

## Data Availability

The data generated in this study are available upon request.

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
