# Peer review of "Tofacitinib Treatment Suppresses CD4+ T-Cell Activation and Th1 Response, Contributing to Protection against Staphylococcal Toxic Shock"

_ijms, 2024, doi:10.3390/ijms25137456_

Round 1
Reviewer 1 Report
Comments and Suggestions for Authors
The manuscript by Jarneborn et al. is well-written, and the experiments are nicely performed. The authors have previously published a similar kind of work in Scientific Reports journal. In the current publication, they have demonstrated that CD4+ T cells are pathogenic in the context of Staphylococcus aureus (SA) toxic shock, which is the primary new experiment compared to their previous work. The remaining data appear to be quite similar to their prior publication.
In this publication, the authors have provided novel insights into the mechanisms by which Tofacitinib treatment contributes to protection against SA toxic shock. This addition of mechanistic details regarding Tofacitinib's protective role adds some element of novelty to the present work.
Minor issue
The authors have not used antibodies against neutrophils and B cells (FITC-conjugated anti-CD69 (Invitro- 300 gen), BV510 anti-CD19 (BioLegend), PE-Cy7-conjugated anti-Ly6G (BD Bioscience), 301 BV605-conjugated anti-Ly6C). Please remove such content from the paper.
Author Response
Comments 1: The authors have not used antibodies against neutrophils and B cells (FITC-conjugated anti-CD69 (Invitro- 300 gen), BV510 anti-CD19 (BioLegend), PE-Cy7-conjugated anti-Ly6G (BD Bioscience), 301 BV605-conjugated anti-Ly6C). Please remove such content from the paper.
Response 1: Thank you very much for pointing this out. It is quite correct that we do explore the role of neutrophils or monocytes in this paper. However, the anti-Ly6G and anti-Ly6C were included in the original panel and while we did not use this data, the antibodies were used to exclude neutrophils (Ly6G+) and monocytes (Ly6C+) and achieve more precise gating of CD3+ cells. So while the neutrophils are not mentioned in the results, we will keep the antibodies in the methods and have them shown in the supplementary gating strategy. We have removed CD19 since it’s quite correct that it was not used for this paper. A legend for supplementary figure 1 of the gating strategy has been added for additional clarification.
Reviewer 2 Report
Comments and Suggestions for Authors
Comments:
The paper titled “Tofacitinib Treatment Suppresses CD4+ T-Cell Activation and Th1 Response, Contributing to Protection Against Staphylococcal Toxic Shock” by Anders et al., the authors explained the effects of tofacitinib on T-cell response in peripheral blood using a mouse model of enterotoxin-induced shock. The author’s observations are interesting but need major revision before the paper gets accepted.
Major Comments:
· It is not clear why the author injected JAK inhibitor Tofacitinib prior infection? Generally, Tofacitinib is used as a therapy. It will be interesting if the authors can infect the mice and do the treatment.
· Fig 1A please check the gating and share the gating strategy in supplementary section.
· The authors show there is an increase in CD3 count post Tofacitinib treatment (Fig.1A) but Fig 1B shows no significant difference it is very confusing.
· It is not clear how the authors check the circulatory T-cell?
· The author shows there is an increase in CD4-T cell post toxin treatment and when they block the CD4-T cell they get only 20% survival. To my understanding 20% survival is not a significant one- and the-time course of study is too small (150 hours). It will be interesting if the authors can use a wide time course for the survival assay.
Author Response
Comments 1: It is not clear why the author injected JAK inhibitor Tofacitinib prior infection? Generally, Tofacitinib is used as a therapy. It will be interesting if the authors can infect the mice and do the treatment.
Response 1: Thank you for this valid observation. We agree that from a clinical stand point it would be highly interesting to treat the mice after exposure to toxins. However, our previous paper showed that tofacitinib injection after toxin challenge did not have any protective effect in this model (Figure 5; Jarneborn et al, Sci Rep 2000, PMCID: PMC7331611) and this is why we chose the pre-treatment design for further exploration with this paper. When treating animals post toxin exposure the results are not as beneficial, so the current study is focused on better understanding the protective effect resulting in possible further understanding on when treatment could be an option.
Comments 2: Fig 1A please check the gating and share the gating strategy in supplementary section.
Response 2: Thank you for your comment. Figure 1A shows representative gates from the three groups of the cell populations deemed most interesting. Supplementary figure 1 shows a representative example from one sample for the full gating strategy. This figure has been updated and a figure legend for the image has been added.
Comments 3: The authors show there is an increase in CD3 count post Tofacitinib treatment (Fig.1A) but Fig 1B shows no significant difference it is very confusing.
Response 3: Thank you very much for this valuable remark. We have tried to make the figure more clear to avoid any risk of miscommunication. Fig 1A shows representative FACS plots from one representative sample in each group. To make it more clear we have changed the tofacitinib sample to one with a lower percentage of CD3+ cells as to not imply a larger difference. Fig 1B shows the summarized graph of all samples with the statistical significance indicated. As shown in Fig 1B there was not significant difference between tofacitinib group and controls or healthy references. With regards to your comment we have changed the statistical method from non-parametric (Kruskal-Wallis) to parametric (one-way ANOVA) after all populations were tested and fulfill normal distribution. This gives statistical difference between both toxin groups accentuating that tofacitinib does not primarily protect by limiting the decrease in CD3+ T-cells. Figure 1 and its legend (page 3, lines 84-93) has been updated accordingly and the results section 2.1, page 2, final paragraph, lines 72-81, has also been adjusted.
Comments 4: It is not clear how the authors check the circulatory T-cell?
Response 4: We refer to T-cells measured in peripheral blood as circulating T-cells as opposed to those resident in lymphatic organs or tissues. The circulating T-cells were analyzed by FACS in peripheral blood.
Comments 5: The author shows there is an increase in CD4-T cell post toxin treatment and when they block the CD4-T cell they get only 20% survival. To my understanding 20% survival is not a significant one- and the-time course of study is too small (150 hours). It will be interesting if the authors can use a wide time course for the survival assay.
Response 5: Thank you for this comment. We agree that 20% is not enough to explain the effect of tofacitinib or to deem CD4+ T-cells as the main culprit behind the pathology of toxin induced shock. However, the result points to CD4+ cells being highly involved since even a 20% reduction is quite strong compared to 100% mortality in other groups, as well as prolonging the time to the first death in CD4-depleted mice. It aligns with the finding that the major difference of the tofacitinib group compared to control group is in fact a lower ratio of activated CD4+ T-cells.
Regarding the time course, 150 h appears as an endpoint for this experiment but the mice who gets though the initial phase of shock recovers fully and live on seemingly unaffected. These survival experiments was ends when all symptoms are gone and the remaining mice are clinically fully recovered for about 72 h. A clarifying sentence has been added to the results section 2.5, page 6, line 158-160.
Round 2
Reviewer 2 Report
Comments and Suggestions for Authors
Comments:
In the revised manuscript “Tofacitinib Treatment Suppresses CD4+ T-Cell Activation and 2 Th1 Response, Contributing to Protection Against Staphylococ- 3 cal Toxic Shock” the authors tries to address the comments but there are still certain issues that need to be clarified.
· In Response 1 as the authors mentioned post injection with tofacitinib after toxin challenge did not have any protective effect, in the present paper pre immunization showed only 20% protection, to my understanding which is not at all significant. The authors should find a different strategy to prove the effect of tofacitinib.
· In the gating strategy it is not clear why the authors selected Ly6C and Ly6G first?
Author Response
|
Comments 1: In Response 1 as the authors mentioned post injection with tofacitinib after toxin challenge did not have any protective effect, in the present paper pre immunization showed only 20% protection, to my understanding which is not at all significant. The authors should find a different strategy to prove the effect of tofacitinib.
|
|
Response 1: Thank you for your insightful comment, which has undoubtedly enhanced the quality of our manuscript. I'd like to provide some additional background information for clarity. In our previous study (Jarneborn et al., Sci Rep 2000, PMCID: PMC7331611), we demonstrated that pre-treatment with tofacitinib offered complete protection against mortality in toxin-induced shock. We observed survival rates between 70-100% in treated mice compared to 0% in untreated mice. It is well-established that T-cell activation plays a crucial role in the pathology of this type of shock, which is why we focused on this cell type in the current study. To verify the importance of T-cell activation and identify which type of T-cell is more pathogenic, we conducted depletion experiments. Our findings showed that depleting CD4+ cells with antibodies (Anti-CD4) before toxin challenge provided a statistically significant protective effect, while depleting CD8+ cells (Anti-CD8) did not. We fully agree with your observation that the 20% survival rate (not 70-100%) in the CD4 depletion group suggests that CD4+ T-cell activation is a significant contributing factor, but other contributors must be acting synergistically with CD4 T cells. For example, TNF is known to be crucial in the enterotoxin-induced model, as TNF Rp55-/- mice are resistant to toxin challenge (PMID: 9130631). Monocytes are the major producers of TNF-alpha, and it's very likely that monocyte activation also contributes to lethal shock. Tofacitinib may inhibit monocyte activation as well. The combined inhibition of CD4 T cells and monocyte activation could explain the full protective effect observed. While our current study focuses primarily on understanding the role of T-cell populations in lethal shock, we plan to explore the role of monocyte activation and its inhibition by tofacitinib in future studies. We have incorporated these points into the discussion section of our manuscript. |
|
Comments 2: In the gating strategy it is not clear why the authors selected Ly6C and Ly6G first? |
|
Response 2: Thank you for your comments. While we do not explore neutrophils or monocytes in this paper, we included anti-Ly6G and anti-Ly6C in the original panel to identify myeloid cells. These antibodies were used to exclude neutrophils (Ly6G+) and monocytes (Ly6C+), allowing for more precise gating of CD3+ cells. A sentence clarifying this has been added to the figure legend of Supplementary Figure 1. |
Round 3
Reviewer 2 Report
Comments and Suggestions for Authors
No Comments. The authors clearly addressed the comments.